# The Relationship between Social Participation and Subjective Well-Being among Older People in the Chinese Culture Context: The Mediating Effect of Reciprocity Beliefs

**DOI:** 10.3390/ijerph192316367

**Published:** 2022-12-06

**Authors:** Xinyu He, Daniel T. L. Shek, Wenbin Du, Yangu Pan, Yin Ma

**Affiliations:** 1Humanities and Law School, Chengdu University of Technology, Chengdu 610059, China; 2Department of Applied Social Sciences, The Hong Kong Polytechnic University, Hong Kong SAR 999077, China; 3Research Institute of Social Development, Southwestern University of Finance and Economics, No. 555, Liutai Avenue, Wenjiang District, Chengdu 611130, China; 4School of Business, Sichuan Normal University, No. 5, Jingan Avenue, Jinjiang District, Chengdu 610101, China

**Keywords:** reciprocity belief, social participation, subjective well-being, mediating effect, Chinese older people

## Abstract

It is demonstrated that the subjective well-being (SWB) of older people greatly relies on their social participation. However, there are few studies on reciprocity beliefs as a mediating mechanism between social participation and SWB. In this study, 297 participants aged 60 and over in Chengdu, Sichuan Province, China, completed a questionnaire of social participation, reciprocity beliefs, and SWB. We used multiple linear regression and mediation analyses to verify the mediating effect of reciprocity beliefs. Results showed that social participation was positively relative to SWB and reciprocity beliefs, and reciprocity beliefs played an intermediary role in social participation and SWB. These findings suggest the importance of social participation for SWB, with reciprocity beliefs (behaviors) playing a positive mediating role, particularly in China. In conclusion, analysis of the mediating effect of reciprocity beliefs provides us with knowledge that could help in achieving a healthy old age. Additionally, this study opens up new perspectives of research.

## 1. Introduction

China’s aging problem is becoming increasingly serious, with 264 million people aged 60 and over, accounting for 18.3% of the total population in 2020. In this sense, China has entered an aging society [1]. Previous literature has shown that with age, the different abilities of people are affected, with a cognitive and physical decline and a reduction in social contact [2,3], which might eventually lead to a decline in SWB [4,5]. Subjective well-being is considered to be people’s affective reactions, domain satisfaction, and life satisfaction [6,7]. Affective reactions include negative and positive emotions [8], which represent a person’s subjective evaluation of the events that happen in their lives. In addition, work satisfaction, family satisfaction, and group satisfaction are three types of domain satisfaction [6]. For older people, the more active they are in the relationships with family, relatives, friends, or other social groups, the more positive emotions they can experience, which is conducive to improving their SWB [9,10]. Additionally, many researchers are interested in life satisfaction, referring to a person’s self-evaluation of his/her life [11]. In gerontology, life satisfaction is a concept often used to assess subjective well-being [12]; for example, a self-evaluation in the past, present, and future [6]. The measurement of subjective well-being may predict the health behavior of the elderly and contribute to the realization of healthy aging [13,14]. As a result, elderly SWB has received increasing attention in recent years [15]. More research into the variables that influence self-evaluation of life satisfaction in older people is of fundamental importance for enhancing their SWB.

Social participation is a form of social interaction, including a series of activities with family, friends, and other individuals [16,17], which has been shown to improve the health and SWB of older adults [18,19,20]. Older people who participate more in social activities have a higher SWB than those who participate less [21,22]. Studies also show older people who spent more time on these social activities could gain a stronger sense of purpose, more resources, and a greater motivation to practice behaviors that improve SWB, all of which may contribute to better well-being [23]. This is, perhaps, because social participation that involves interaction or communication with others provides immediate emotional benefits for older people and enhances their sense of social support [24]. Olesen and Berry conducted interviews with eight elderly people, who narrated their experiences of participating in social activities, and results showed that social participation is beneficial to improving self-efficacy, a sense of belonging, and mental health [25].

To our best knowledge, in China, social communication that has developed over thousands of years of agricultural civilization has its roots in the small, localized peasant economy, which emphasizes human relationships between family members, relatives, and neighbors [26]. The interaction with family, friends, and neighbors is an important factor that can increase positive emotions and improve SWB among older Chinese people [27,28,29].

Activity theory supports the correlation between social participation and SWB noted above. Activity theory suggests that older people who regularly participate in social activities may maintain a positive self-concept and improve their sense of accomplishment, both of which contribute to increased well-being in later life [16]. This theory is often used to explain the link between social participation and well-being in older people. Active participation in social activities can expand one’s social network, because, after leaving the workplace, many older adults lose the social ties that younger people develop through work [30]; thus, social participation can not only strengthen social ties, but also increase the exchange of information and emotional support in the elderly, and, thereby, further enhance SWB [31,32,33].

Another problem highlighted in the literature is that the determinants of life satisfaction may differ by gender and cultural context [34,35]. Studies show that in terms of gender, there is different SWB for men and women [36,37]. Okabayashi and Hougham found there were some gender differences in the influence of social interaction on the happiness of the elderly [38]. Older men tend to engage in going to the bar and going out outdoors, while for older women, travel, sports, or tourism are the social participation activities that predict SWB [39].

In addition, empirical research has determined that reciprocity beliefs are associated with both social participation and SWB in older people [40,41,42,43]. Reciprocity beliefs refer to individual’s idea of reciprocity-based behavior and their expectations that others will engage in such behavior [44]. Reciprocity beliefs have held a foundational place in Chinese culture since ancient times [45]. The traditional Chinese mentality is based on a familial and social construction of *wenqing* (warm-heartedness), through which everyone can receive and pay back material resources and spiritual wealth. *The Book of Songs*, written in the middle of the Spring and Autumn period in Chinese history (roughly 771 to 476 before the Christian Era), contained the idiom: “Give me a peach and I will give you a plum.” [46,47]. These form the foundation of a strong sense of reciprocity among Chinese people, who typically reestablish networks of relationships and maintain long-term communication with others through social participation, mutual assistance, and reciprocity [48,49]. Therefore, in the context of Chinese culture, reciprocal relationships with others are an important factor in their lives [50]. Moreover, social participation is the basis and premise for the conversion of reciprocity beliefs into reciprocity behavior. Repeated social interactions that reflect expectations of reciprocal exchange can reinforce reciprocity beliefs and drive the mutualistic escalation in the relationships [51]. Older people with extensive social participation and strong beliefs in reciprocity, who exert their proactivity in social interactions, experience more positive affect and receive more empathy from others, which in turn translates into higher SWB [52,53].

According to the socioemotional selectivity theory [54], those who grow older are more likely to choose social participation that provides them with close social relations. Thus, compared with other groups, older people tend to choose social networks with strong interactive relationships and stronger reciprocity beliefs, and the regard the principle of reciprocity as a way to maintain SWB. Research on parent-child relationships between generations indicate that the elderly parents in families who had reciprocity beliefs reported higher happiness than those who did not hold such reciprocity beliefs [55]. Other studies have demonstrated that reciprocity beliefs could promote mental well-being in older people [56]. In a study involving 244 Chinese adolescents, results showed that positive reciprocity beliefs play a moderating role in the relationship between the development of a proactive personality and personal life satisfaction [42]. In another study, results showed that more positive perceptions of balanced relationships, equitable treatment, and reciprocity in the interpersonal domain were associated with higher SWB [56]. A negative correlation between reciprocity beliefs and SWB has also been revealed in research, which found that the less reciprocity beliefs, the greater well-being [57].

As the existing research is not consistent, it would be useful to further examine the relationship between reciprocity beliefs and SWB. It is worth noting that, in the Chinese culture context, where the expectation that “the courtesy demands reciprocity” is highly valued [58], it could be particularly fruitful. The Confucian concept of ren’ai (seek also to establish others if wishing to be established oneself; seek also to enlarge others if wishing to be enlarged oneself) has a strong influence on Chinese society as a value that stabilizes the social order and maintains harmonious communication among Chinese people [59]. One particularly important feature of ren’ai, helping behavior (e.g., volunteering in the community, providing emotional support), has long been found to increase the well-being of the elderly [60]. In addition to ren’ai, Chinese people also focus on bao’en (when you drink the water, remember its origin). In such an ethically oriented society, the expectation is that Chinese people have a deeply-held debt-paying mentality combined with a cultural aim to seek harmony and balance [61]. When they engage in relationships with others, those who receive help have obligations to reciprocate, and such reciprocation may be more instrumental, such as solving problems or sharing responsibilities [53]. In a sense, Confucianism is a way of life that offers a theory about interpersonal relationships that holds that a harmonious relationship can only be realized in the context of human reciprocity [62].

The fact that the relationship between social participation and SWB could be mediated by reciprocity beliefs can be explained by the reciprocity theory of exchange [63]. According to this theory, reciprocity is structured with respect to the way the expected behaviors and benefits of exchange are connected, thus different reciprocal structures involve different forms of exchange [64]. People communicate and connect with each other through social participation and establish social relationships [65], and, in particular, reciprocal relationships are believed to improve SWB in older people [66]. In addition, the purpose of the social participation of older people is not to obtain certain benefits; there is more uncertainty about giving and giving back, which are more likely to seem voluntary [67]. In contrast, people give help, resources, and affection to others willingly and with kind intent [68]. The strong belief in reciprocity held by Chinese people influences the way they engage in exchange in relationships. In the reciprocal process of getting help or giving help to others, the older people of any culture are more likely to experience a sense of satisfaction and achievement, which is conducive to improving their SWB.

In sum, Chinese people pay attention to the social communication mode of “reciprocity”, and the social participation of older people is both an important way to establish communication with others and the premise and basis for the practice of reciprocity beliefs. Older people with strong reciprocity beliefs can gain more instrumental and emotional support to improve their SWB through active participation in social activities [56].

Although these associations are well known, few research studies have included all these variables, and, in particular, the path from social participation to SWB via reciprocity beliefs in older people has not been fully demonstrated. In the context of Chinese culture, this study explored the mediating role of reciprocity beliefs between the social participation and SWB of older people. Therefore, we propose the following four hypotheses:

**Hypothesis** **1.***Social participation is positively related to SWB in older men and women*.

**Hypothesis** **2.***Social participation is positively related to reciprocity beliefs in older men and women*.

**Hypothesis** **3.***Reciprocity beliefs are positively related to SWB in older men and women*.

**Hypothesis** **4.***Reciprocity beliefs play a mediating role in social participation and SWB in older men and women*.

## 2. Materials and Methods

### 2.1. Participants

This study was conducted between 20 December 2016 and 25 January 2017 in five districts in Chengdu: Jinjiang, Wuhou, Qingyang, Chenghua, and Jinniu Districts. The study included people from 15 communities. The participants were aged 60 and over, and they lived in the above communities. The communities in each administrative region were numbered one by one in alphabetical order, and the communities numbered 01, 11, and 21 in each administrative region were selected, respectively, and a random sample of 20 people in each community and 60 people in each administrative region were surveyed. A total of 300 people in 5 administrative regions were selected for the questionnaire survey, and 297 valid questionnaires were collected. Finally, 297 Chinese people aged 60 and above, with help and explanation as needed from professional researchers, completed the survey that formed the basis of this study. The survey included socio-demographic information, as well as social participation, reciprocity beliefs, and SWB variables.

### 2.2. Measures

#### 2.2.1. Subjective Well-Being (SWB)

In this study, SWB was measured following a similar approach to that taken by Cui [69], who developed 20 items based on Diener’s concept of life evaluation (past, present, future) to measure the SWB of older people. Six questions were used to measure past SWB, for instance, “There were few regrets in my life”; “Compared with my peers, I had done a lot of stupid things”; “Looking back on my life, I had done almost nothing right”; “My life was very hard and lonely, I do not want to think about it”. Eight questions were used to measure present SWB, for example, “Recently, I feel very happy in my life”; “I am happy with my lifestyle”; “Everyday, life is very boring”; “Recently, I am prone to get angry”; “Recently, there are more sad things than happy things”. Six questions were used to measure future SWB, such as “I am full of hope for the future”; “In the future, there will be many interesting and happy things around me”; “In the future, there is almost nothing that I will be able to do”; “In the future, I will no longer be able to do anything of value”. Statements were responded to on a five-point frequency scale ranging from 1 (never), 2 (seldom), 3 (sometimes), 4 (regularly), to 5 (very frequently), and responses were partially reverse-coded so that the higher score indicates greater SWB. The Cronbach’s alpha in the present study is 0.821.

#### 2.2.2. Social Participation

Social participation was measured following a similar approach to that taken by Song [70], who developed 28 items to measure the social participation of older people (including participation in family gatherings, hobby groups, religious groups, political groups, etc.). According to the design of this article and Chinese cultural background, five questions were used to measure participants’ involvement in the following participation. For example, “Do you participate in community voting activities?”; “Do you get together with family, friends, and neighbors, to go shopping or travel?”. All of the items were rated using the aforementioned five-point frequency response scale. The Cronbach’s alpha in the present study is 0.663.

#### 2.2.3. Reciprocity Beliefs

Reciprocity beliefs were measured following a similar approach to that taken by Lin [71], who used 7 items to measure the general rules and reciprocity rules of older people. According to the design of this article, we adopted four questions to measure the reciprocity beliefs. For example, “Have you helped others?”; “Do you believe that, when you need help, the people who will help you will be those who received help from you previously?”; “Have you received help from others?”; “Will you help others who helped you previously?”. The questions were answered on a five-point-scale ranging from 1 (completely do not believe), 2 (believe less), 3 (does not matter), 4 (believe more), to 5 (completely believe). The Cronbach’s alpha in the present study is 0.678.

#### 2.2.4. Demographic Measures

We collected demographic variables of gender (0 = male; 1 = female), age (0 = 60–69; 1 = 70–79; 2 = 80 and above), educational attainment (0 = technical college and above; 1= primary school and below; 2 = middle and high school), self-rated health status (0 = healthy; 1 = average; 2 = unhealthy), whether living with children (0 = live with children; 1 = do not live with children; 2 = no children), income status (0 = ≤3000 RMB; 1 = 3001–4999 RMB; 2 = ≥5000 RMB), and self-rated socioeconomic status (0 = higher; 1 = average; 2 = lower).

### 2.3. Data Analytic Strategy

We used the regression model for data analysis in SPSS22.0. First, we performed a descriptive analysis of all variables (see Table 1). Second, we used correlational analyses to test whether reciprocity beliefs were associated with the mediators and outcome variable in the expected directions (see Table 2 and Table 3). Third, we used multiple mediation analysis to verify the relationship between social participation, SWB, and reciprocity beliefs (Hypothesis 1, 2, and 3; see Table 4 and Table 5; Figure 1 and Figure 2). Finally, in order to examine whether there were mediation effects of reciprocity beliefs between social participation and SWB, we used the SPSS Macro PROCESS program 4.1 designed by Preacher and Hayes [72] to complete data analyses. The bootstrap confidence interval (CI) was set to 95%, and the number of bootstrap samples was 5000. If the CI of the indirect effect did not include zero, the mediation effect was proved to be significant (Hypothesis 4; see Table 6 and Table 7).

## 3. Results

### 3.1. Descriptive Results

Table 1 shows the descriptive statistics of the variables. Though the average older men and women scores with respect to SWB were similar, above neutral (3.0), and, thus, leaning towards the “satisfied” side, the average SWB levels of men (=3.74) were slightly higher than those of women.

Among the age groups, men aged 70 to 79 accounted for the largest proportion, while women aged 60 to 69 accounted for the largest proportion. As to educational attainment, compared to women, older men had attained a higher rate of college education, with around 26.9% being educated at that level. More than half of the older people considered themselves to be of average health, with only about 10% acknowledging that they are unhealthy. Even though the older people surveyed in Chengdu indicated that they do not want to live with their children, older women were more likely to express a desire to live (or consider living) with their children than their male counterparts. The monthly income of most of the older people in this study is below RMB 3000. Compared to the older women in this study, a higher proportion of the older men had a monthly income over RMB 5000. Regardless of gender, no older people participants considered themselves to have a high socio-economic status.

### 3.2. Correlation of Social Participation, Reciprocity Beliefs, and SWB

The correlations between the examined variables are shown in Table 2 (older men) and Table 3 (older women). The results of the correlation analyses are statistically significant (*p* < 0.001). First, social participation is significantly positively correlated with reciprocity beliefs (*r*_1_ = 0.429; *r*_2_ = 0.371). Second, social participation is significantly positively correlated with SWB (*r*_1_ = 0.515; *r*_2_ = 0.418). Third, reciprocity beliefs are significantly positively correlated with SWB (*r*_1_ = 0.638; *r*_2_ = 0.501).

### 3.3. Relationship between Social Participation, Reciprocity Beliefs, and SWB

The multiple mediation analyses are presented in Table 4 (older men) and Table 5 (older women). The results showed that, when controlling for age, educational attainment, self-rated health status, whether living with children, income status, and self-rated socioeconomic status, social participation is positively related to reciprocity beliefs (a_1_ = 0.401, a_2_ = 0.329, *p* < 0.001). In addition, reciprocity beliefs are also positively related to SWB (b_1_ = 0.317, b_2_ = 0.377, *p* < 0.001), and social participation is positively related to SWB (c’_1_ = 0.425, c’_2_ = 0.419, *p* < 0.001) (see Figure 1 and Figure 2).

### 3.4. Mediation Effect of Reciprocity Beliefs on the Relationship between Social Participation and SWB

The mediation analysis found that the relationship between social participation and SWB is partially mediated by reciprocity beliefs in older men and women (see Figure 1 and Figure 2, Table 6 and Table 7), the direct effect of social participation on SWB remained significant in older men and women (Β_1_ = 0.153, *p* < 0.01; Β_2_ = 0.166, *p* < 0.001), and the indirect effects of social participation on SWB through reciprocity beliefs are significant in older men and women (indirect effects = 0.086 and 0.054; Bootstrap 95% CI = [0.039, 0.155] and [0.024, 0.094], respectively). Moreover, the contrast test of the indirect effects found that the indirect effect of reciprocity beliefs was stronger in older men (36%) than in older women (25%).

## 4. Discussion

Our research used the self-evaluation of life satisfaction to assess subjective well-being among older people. The results suggest that older people with frequent social participation have higher SWB and stronger reciprocity beliefs. These results are consistent with previous studies that show older people who have frequent social participation can obtain more social resources and, thus, improve their SWB [73,74]. Finding social participation to be positively related to reciprocity beliefs could be explained in the context of the socioemotional selectivity theory [54]; that is, with age, older people tend to choose those social relationships that can provide them social reciprocity, which can enhance their health and happiness in China, where reciprocity beliefs are widely recognized and valued [75]. In societies, older people are also expected to play a valuable role and social interaction is reciprocal, which is a guiding principle for maintaining harmony [76]. Thus, in the Chinese cultural context, social participation is a key way for older people to establish social relations after retirement, and it is the carrier of the beliefs of practical reciprocity.

We also found reciprocity beliefs were positively related to SWB in older men and women. This is similar to the previous studies. Tuominen and Haanpää‘s study explored the association between the social capital of young people and their SWB using Finland’s sub-sample of the third wave of the International Survey of Children’s Well-Being. Their results showed that reciprocity is related to SWB [77]. Another study showed that the importance of perceived reciprocity in SWB increases with age [78]. According to that study, to improve the mental health of the elderly, reciprocity in social exchanges is an important factor [66], and it should be noted that older Chinese people have reported having more reciprocal friendships of emotional solace than young Chinese people have [56]. People who hold reciprocity beliefs know that reciprocal social interactions are beneficial for both the group and the individual; thus, they display positive giving behaviors to their peers, showing more sincere care and help towards others. Those who receive the help have a willingness to repay, such as solving problems, thereby sharing the pressure. In addition, people who have strong beliefs in reciprocity may receive more emotional comfort, such as being heard, receiving affective rewards, and sharing experiences. These emotional feedbacks fulfill the need for belonging, making older people more satisfied with their lives [53]. Aging is inevitable, and most of us experience gradual decline in physical function during this period, resulting in an increased need for help from others. Rather than paying attention to the negative dependency of older people associated with decline and disengagement, interdependence may be constructed as a phase of reciprocity and connectedness, i.e., seeking and accepting help that can be viewed as a way to maintain connectedness between people [79]. As self-determination theory suggests [80], life satisfaction of basic need or emotional need, such as the need for autonomy, competence, and social connection, enhances individual happiness.

Our results found that reciprocity beliefs mediate the effect of social relationships on SWB in older people. This result is consistent with the finding of Wahrendorf’s study [81] that, based on reciprocity in the activities of older adults in mainland Europe, participation in socially productive activities was associated with happiness. Furthermore, Mcmunn’s study underscored that the reciprocity of social relations can be used to explain the relationship between participation in socially productive activities and the well-being of post-retirement age people in England [82]. It has been argued that social participation provides immediate emotional comforts through interaction or communication with others and expands one’s social support [24]. Chinese people value harmony and balance, which is achieved through the practice of reciprocity beliefs. Of course, Western cultures also value reciprocity, but such tendencies may be more strongly imposed in Chinese society [49]. The ancient Chinese book *The «Guodian Bamboo Slips»* states, “Dao begins with feelings”. This means that relationships between people originate from feelings. A unique collectivist, cultural context in Chinese societies promotes the value of social relationships and connectedness with others, including families, friends, and neighbors [83].

Finally, this study also revealed the mediating effects of reciprocity beliefs are stronger in older men than older women. This is in agreement with Tsai and Dzorgbo’s study, who found gender differences have substantial impacts on reciprocity and an individual’s SWB; compared to women, Ghanaian men are more likely to maintain reciprocity in later life [84]. However, a study shows that, in Europe, ‘reciprocal-like’ giving and receiving is more likely to occur and be expected among elderly women and those with a network of relationships within close distances [85]. This suggests that reciprocity beliefs may be generally stronger among older women. According to our results, we predicted that reciprocity beliefs would have a greater impact on older men’s SWB. Since, during old age, the resources owned further decline [86], it may be increasingly difficult for older people to return the support received, making older people less likely to maintain reciprocal relationships in old age [87]. However, the value of reciprocity beliefs is deeply ingrained in Chinese people [88], who want to maintain long-term relationships through reciprocity. According to the Chinese retirement scheme, the retirement age is 60 for men, 55 years for female professionals/cadres (including teachers, administrators medical personnel, and other personnel of public institutions), and 50 years for other female workers. As, in China, women often retire earlier than men, they have the ability to adapt to retirement earlier, and they also have more time and opportunities to participate in social activities. In Chinese tradition, older women are usually not expected to work for a living, and they prefer to participate in many social activities [89]. They tend to interact with and support each other through social participation, even if there is no explicit exchange purpose. Consequently, compared to older men, older women have more opportunities to form reciprocal relationships with others in China. According to this view, since older men are more threatened by the goal of reciprocity relationships, they may try harder to achieve it [77]; hence, the mediating effects of reciprocity beliefs are stronger.

Indeed, improving SWB is beneficial to the physical and mental health of the elderly and reduces depression [90,91]. How to enhance the SWB of older people is an important social issue. Therefore, in order to clarify the factors affecting SWB, we urge researchers not to limit future studies to the direct effects of social participation on SWB in older people. We call for policymakers to encourage older people to participate in neighborhood or community affairs in China, thereby promoting the positive identity older people could achieve through social networks of community inclusion, interdependence and reciprocity, and exploring available exchange resources. Determining how best to assist older people’s recognition of the value of their existing exchange resources in reciprocal relationships, as well as fostering meaningful interactions among participants, is likely to be important [92]. Although older people may not join a given social activity with the intention of exchanging benefits, engaging in activities that stimulate reciprocity can facilitate greater overall social participation levels. In fact, the emotional exchange of human contact is a kind of reciprocity that can produce strong bonds of trust and solidarity between actors; in turn, these bonds will improve the enthusiasm of the participants to continue to engage in social activities [64]. One way to do this would be to reciprocate the involvement of older people as participants or volunteers in a community cultural or sports activity by offering them practical support, such as long-term care insurance or other forms of financial and social assistance, in return.

## 5. Limitations and Future Directions

Limitations of this study are as follows. First, since the data are cross-sectional and lack longitudinal data, it is difficult to show changes in social participation, subjective well-being, or reciprocity beliefs. By collecting long-term data, a future study could examine longitudinal changes and causality. Second, although this study emphasized the importance of Chinese reciprocity beliefs, it was unable to take into account the different expectations of family members and acquaintances when it comes to reciprocity beliefs in China. To explore the impact of different relationships on SWB more precisely, reciprocity beliefs could be further subdivided into reciprocity with “acquaintances” and reciprocity with “strangers” in future research. Finally, the data was limited to the population surveyed: elderly males and females from Chengdu City, Sichuan Province. Given that China is a vast country, and its regions have unique demographic characteristics, it would be interesting to investigate any variations between urban and rural areas, and between different provinces.

## 6. Conclusions

This study examined the associations between social participation and SWB and the mediation effects of reciprocity beliefs between social participation and SWB among a sample of older Chinese adults. Our research results confirmed all the hypotheses we made in the paper. We found that social participation was associated with both SWB and reciprocity beliefs, and we also found reciprocity beliefs could mediate the relationship between social participation and SWB in older people in China. These findings suggest the importance of social participation for SWB, with reciprocity beliefs (behaviors) playing a positive mediating role, particularly in China. In conclusion, analysis of the mediating effect of reciprocity beliefs provides us with knowledge that could help in achieving a healthy old age. Additionally, this study opens up new perspectives for research.

## Figures and Tables

**Figure 1 ijerph-19-16367-f001:**
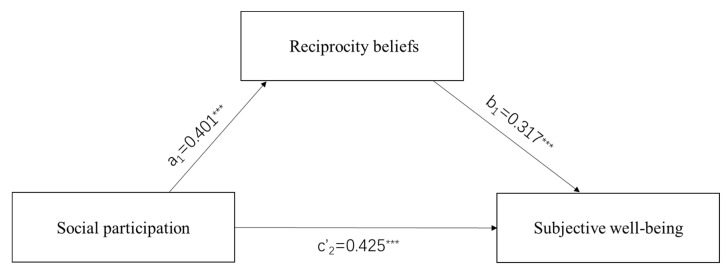
Mediation model of older men in Chengdu. *** *p* < 0.001.

**Figure 2 ijerph-19-16367-f002:**
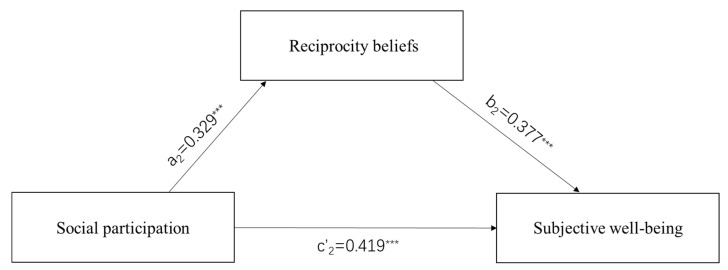
Mediation model of older women in Chengdu. *** *p* < 0.001.

**Table 1 ijerph-19-16367-t001:** Descriptive statistics of variables.

Variables	Percentage/Mean Value (Standard Deviation) (%)	
	Men *n* = 131	Women *n* = 166
SWB	3.74	3.70
Age		
60–69	36.7	44.0
70–79	45.1	41.0
80 or above	18.2	15.1
Educational attainment		
Primary school and below	35.7	41.6
Middle and high school	37.4	35.5
College education or above	26.9	22.9
Self-rated health status		
Unhealthy	10.4	12.7
Average	53.5	51.8
Healthy	36.0	35.5
Whether living with children or not		
Yes	34.0	41.0
No	64.6	57.2
No children	1.3	1.8
Income status		
≤3000 RMB	70.7	80.7
3001–4999 RMB	22.2	15.1
≥5000 RMB	7.1	4.2
Self-rated socioeconomic status		
Lower	48.5	50.0
Average	51.5	50.0
Higher		

**Table 2 ijerph-19-16367-t002:** Correlations between social participation, reciprocity beliefs, and SWB of older men.

Variables	Social Participation	Reciprocity Beliefs	SWB
Social participation	-		
Reciprocity beliefs	0.429 ***	-	
SWB	0.515 ***	0.638 ***	-

*** *p* < 0.001.

**Table 3 ijerph-19-16367-t003:** Correlations between social participation, reciprocity beliefs, and SWB of older women.

Variables	Social Participation	Reciprocity Beliefs	SWB
Social participation	-		
Reciprocity beliefs	0.371 ***	-	
SWB	0.418 ***	0.501 ***	-

*** *p* < 0.001.

**Table 4 ijerph-19-16367-t004:** Multiple linear regression analysis results in older men.

Variables	Reciprocity Beliefs		SWB		
	Model 1	Model 2	Model 1	Model 2	Model 3
Age	−0.155	−0.181 *	−0.137	−0.134	0.134 *
Educational attainment	−0.224 *	−0.257 *	−0.063	0.003	0.022
Self-rated health status	0.070	0.011	0.201 *	0.149 *	0.134 *
Whether living with children or not	0.010	−0.027	0.064	0.093	0.020
Income status	0.126	0.155	0.113	0.073	0.105
Self-rated socioeconomic	0.184 *	0.091	0.232 *	0.158 *	0.105
Social participation		0.401 ***		0.425 ***	0.320 **
Reciprocity beliefs					0.317 ***
SWB					
R^2^	0.052	0.199	0.099	0.280	0.357
∆R^2^	0.095 *	0.147	0.140	0.152	0.078
F	2.182 *	5.611 ***	3.368 **	10.167 ***	12.471 ***
∆F	2.182 *	23.780 ***	3.368 **	34.901 ***	20.030 ***

* *p* < 0.05, ** *p* < 0.01, *** *p* < 0.001.

**Table 5 ijerph-19-16367-t005:** Multiple linear regression analysis results in older women.

Variables	Reciprocity Beliefs		SWB		
	Model 1	Model 2	Model 1	Model 2	Model 3
Age	−0.08	−0.001	0.033	−0.164 *	−0.096
Educational attainment	−0.083	−0.062	−0.025	−0.097	0.000
Self-rated health status	0.109	0.048	0.228 **	0.139	0.135
Whether living with children or not	0.252 **	0.230 **	0.121	0.026	0.036
Income status	−0.031	−0.101	0.163	0.143	0.085
Self-rated socioeconomic	0.191 *	0.167 *	0.188 *	0.135	0.101
Social participation		0.329 ***		0.419 ***	0.268 **
Reciprocity beliefs					0.377 ***
SWB					
R^2^	0.101	0.191	0.126	0.260	0.369
∆R^2^	0.134	0.091	0.158	0.160 ***	0.108 ***
F	4.088 **	6.548 ***	4.982 ***	7.530 ***	10.494 ***
∆F	4.088 **	18.597 ***	4.982 ***	28.086 ***	22.173 ***

* *p* < 0.05, ***p* < 0.01, ****p* < 0.001.

**Table 6 ijerph-19-16367-t006:** Summary of mediating results of older men.

95% Confidence Interval					
Variables	Effect	SE	*p* Value	LLCI	ULCI
Indirect effect					
SP→RB→SWB	0.086	0.030		0.039	0.155
Direct effect					
SP→SWB	0.153	0.046	0.001	0.063	0.243
Total effect					
SP→SWB	0.239	0.045	0.000	0.150	0.329

Abbreviations: SP, social participation; RB, reciprocity beliefs; SWB, subjective well-being.

**Table 7 ijerph-19-16367-t007:** Summary of mediating results of older women.

95% Confidence Interval					
Variables	Effect	SE	*p* Value	LLCI	ULCI
Indirect effect					
SP→RB→SWB	0.054	0.018		0.024	0.094
Direct effect					
SP→SWB	0.166	0.037	0.000	0.093	0.240
Total effect					
SP→SWB	0.220	0.037	0.000	0.150	0.329

Abbreviations: SP, social participation; RB, reciprocity beliefs; SWB, subjective well-being.

## Data Availability

Data are available from the authors. Any interests, please contact the author of correspondence.

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
