# Peer review of "The Relationship between Social Participation and Subjective Well-Being among Older People in the Chinese Culture Context: The Mediating Effect of Reciprocity Beliefs"

_ijerph, 2022, doi:10.3390/ijerph192316367_

Round 1

Reviewer 1 Report

This study examines the relationships between social participation and the subjective well-being of older people in China, as well as the role of reciprocity beliefs in these relationships. The authors do a good job of reviewing the literature, which helps to contextualize and justify the study, although there are aspects that should be expanded. Likewise, the methodology seems adequate, but further clarification is needed on certain aspects that need to be clarified. In general, the results may have very important implications for the improvement of the quality of life of elderly people in China through their social participation, so the article is an interesting contribution as long as some theoretical and methodological aspects are clarified. In any case, I would like to congratulate the authors for their work.

Introduction

I consider that the authors have made a good introduction, using a wide and varied literature that gives a theoretical background to all the ideas presented. Nevertheless, I believe that an effort should be made to broaden the explanation of some key constructs and thus improve the theoretical justification of the research. In addition, authors should make an effort to reorganize the text taking into account the standards of the journal. Here are some comments that may help them.

First, I believe that the initial lines should be rewritten. The first sentence does not use punctuation marks properly: it would be more appropriate to put a period after 2020 and, after the period, write something like: “In this sense, China has entered an ageing society”. Similarly, the second sentence is meaningless: what about cognitive and physical impairments and reduced social contact? I think there is a missing word to indicate that these aspects increase. In addition, I believe that this second sentence, which is very important in order to specify the subject of the work, needs a more general perspective. For example: "With age, the different abilities of people are affected, with an increase in cognitive and physical deficiencies and a reduction in social contact...".

On the other hand, SWB is defined as the cognitive assessment of a person's happiness with life and affective reactions in all areas of his or her life. What are these areas? It should be specified to subsequently better understand how social participation or reciprocal beliefs may affect this well-being. For this purpose, you should make use of the work of Diener, one of the great experts in the field. In fact, the reference by Diener et al. (1999) that you use is very useful in this regard. Authors only make a brief mention of positive/negative emotions, but it is not made clear whether these are part of the SWB. It is necessary to clarify the four main components of this construct: pleasant affect, unpleasant affect, life satisfaction and domain satisfaction. Several of them are closely related to social issues, so this may help the reader to understand certain relationships between variables (satisfaction with work, family and group within the domain satisfactions component, or others' view of the individual's life within the life satisfaction component). I consider it key to improve the quality of the manuscript to expand the theoretical background on SWB.

Likewise, to give more relevance to the study and justify it in terms of its implications for the health of elderly people in China, the influence of SWB on other domains of people's well-being, such as physical or mental health, should be discussed.

I would like to congratulate the authors for their great work in contextualizing the whole issue of social interaction in Chinese society from line 42 onwards. As a minor comment, I would advise them to consider changing the expression time immemorial (line 55) for a more formal one, such as "since ancient times" or "since the beginning of Chinese civilization".

On the other hand, I recommend the authors to check the standards of the journal and make sure of the sections that the manuscript should contain. The section they refer to as Literature Review is not covered, so this information should be included in the introduction. To facilitate the reader's understanding, it may be useful to use the same subsections into which you have divided the information from the literature review. I would recommend that they try to integrate it with the text on lines 42-72, where the different variables and the associations between them are explained in summary form. For example, section 2.1 (now 1.1) could be integrated into lines 42-52. Sections 2.2 (now 1.2) and 2.3 (now 1.3) could be integrated into lines 53-72 and, finally, create a section to discuss the mediating role of reciprocal beliefs in the relationship between social participation and SWB.

Likewise, the objective stated in lines 75-77 would be better placed next to the hypotheses, to facilitate the understanding of both the hypotheses themselves and the method subsequently employed.

Materials and method

The information presented by the authors in this section is adequate, but very important information is missing.

I believe it is appropriate to follow a logical order to present the information. First, lines 187-188 are more specific to the procedure than to the participants. In this regard, I suggest that a first section on "Study design and procedure" be developed at length. In this section they must state the type of design used, which is not clearly specified. Likewise, information should appear about where and when the study was carried out, how the sample was accessed and the information was collected, whether data were eliminated from participants who did not meet the inclusion criteria, etc.

After this, it is more logical to present information about the sample, such as the number of participants, their age and gender. In order for the sample to be sufficiently described in this section, it may be interesting to include Table 1 with detailed sociodemographic information. It may also be useful to specify in this section what were the inclusion criteria for participation in the study.

The measures used are described extensively and information on their psychometric properties is presented for the present sample. However, I am concerned about the validity of the measures. Have validated full scales been used? If so, they should specify the name of these scales and the psychometric properties of the original version. If, on the other hand, only some items of these scales have been used, it should be justified why those specific items are used. Also, does the instrument used to measure SWB measure all the components mentioned by Diener et al. (1999)? If this is not the case, then it should be justified whether this instrument really measures all these components or only focuses on the emotional aspects. In any case, it is important to justify that this instrument is valid and reliable for the measurement of SWB.

The data analyses are adequate and perfectly described.

Results

The results are clearly shown and correspond to the analyses proposed in the previous section.

Only, I recommend the authors not to mention the fulfillment of the hypotheses in the results (e.g. lines 269 and 300).

Discussion

The discussion is adequate, since it is based on the results obtained and is supported by previous literature. Only a few recommendations:

In line 327 the authors say that this is similar to previous studies, but they only cite one study. I think it would be better if they briefly explain what this study (71) is about in order to make it more easily comparable.

On the other hand, in the introduction I recommended the authors to mention all the components of the SWB. In this sense, it would be appropriate to mention them here also on the basis of the results obtained. If the study only yields information about the emotional aspect of SWB, this should be specified so as not to exceed the true scope of the results obtained.

In addition, a large part of the text is devoted to discussing the differences between men and women in different research studies similar to this one. If so much importance is going to be given to this, information in this regard should be set out in the introduction, so that the discussion of this topic makes more sense.

Finally, the results show very positive data on the importance of social participation for the well-being of the elderly. This has major implications that are not sufficiently emphasized. I believe that the authors should use their results to highlight the true potential of their research: the importance of promoting social participation plans for the elderly as a way to improve their quality of life. In this regard, I recommend that they bring to the discussion information about the impact that subjective well-being may have on other spheres of well-being and health of these individuals.

Reviewer 2 Report

The title is adequate to the research problem being undertaken. The paper has been correctly divided into relevant sections, and their content coincides with their titles.

Most of the literature items presented in the reference list are current. They are all related to the topic presented. The number of references is satisfactory and gives a picture of the presented topic

Footnotes and bibliography are in my opinion correctly formulated.

The correct terminology was used. The language of the article is correct, adequate.

The authors proposed four research hypotheses. However, I did not find the purpose of the article.

Research hypotheses was given. Authors should refer to them at the end of the paper.

The conducted research provides grounds for interesting conclusions

Conclusions part should be extended.

The technical part of the article does not raise any objections. The work is aesthetic.

Overall, I rate the paper as very well prepared.

Round 2

Reviewer 1 Report

Thanks to the authors for having taken my proposals into account and congratulations for the significant improvement in the quality of the manuscript. Here are some brief comments that I believe may help to give the manuscript the final quality that this interesting study deserves.

In lines 42-43, I would not repeat the term life satisfaction twice. I think this option would be better: Also, many researchers are interested in life satisfaction, referring to a person’s self-evaluation of his/her life.

I recommend that the authors review the punctuation marks. For example, in line 139, after the parenthesis there should be a period, not a comma. Another example is in line 140, where there is a space between "that" and the comma that should not exist. Also, I would recommend that you put a period between the hypothesis number and the text of what the hypothesis consists of (i.e., Hypothesis 1. Social participation...).

I still miss some aspects of the instruments used. What scales have been used to measure social participation and reciprocity beliefs? Who are the authors and what psychometric properties did they establish in the original version? You give examples of the questions used, but do they form a scale? Are they single questions? Have only a few items been selected from a larger scale? Failure to specify these aspects may cast doubt on the quality of the measures used.

Likewise, in social participation you measure formal and informal participation. This should be justified in the introduction, it is not clear what each represents.

Author Response

First of all, we would like to express our sincere gratitude to the reviewers for their constructive and positive comments.

In lines 42-43, I would not repeat the term life satisfaction twice. I think this option would be better: Also, many researchers are interested in life satisfaction, referring to a person’s self-evaluation of his/her life.

Response: Thank you for your insightful suggestion. According to your opinion, we have revised these two sentences (Page 1, Lines 42-44).

I recommend that the authors review the punctuation marks. For example, in line 133, after the parenthesis there should be a period, not a comma. Another example is in line 134, where there is a space between "that" and the comma that should not exist. Also, I would recommend that you put a period between the hypothesis number and the text of what the hypothesis consists of (i.e., Hypothesis 1. Social participation...).

Response: Thank you for your insightful suggestion. According to your opinion, we have revised the punctuation marks (Page 3, Line 133 and 134; Page 4, Line 167, 168, 170 and 171).

I still miss some aspects of the instruments used. What scales have been used to measure social participation and reciprocity beliefs? Who are the authors and what psychometric properties did they establish in the original version? You give examples of the questions used, but do they form a scale? Are they single questions? Have only a few items been selected from a larger scale? Failure to specify these aspects may cast doubt on the quality of the measures used.

Response: Thank you for your insightful suggestion. According to your opinion, we added the content of measure scales (Page 5, Lines 207-215 and 217-220).

Likewise, in social participation you measure formal and informal participation. This should be justified in the introduction, it is not clear what each represents.

Response: Thank you for your insightful suggestion. According to your opinion, we have deleted the sentence with unclear meaning.